# Deep Ultraviolet Light-Emitting Diode Light Therapy for *Fusobacterium nucleatum*

**DOI:** 10.3390/microorganisms9020430

**Published:** 2021-02-19

**Authors:** Soichiro Fukuda, Shunsuke Ito, Jun Nishikawa, Tatsuya Takagi, Naoto Kubota, Ken-ichiro Otsuyama, Hidehiro Tsuneoka, Junzo Nojima, Koji Harada, Katsuaki Mishima, Yutaka Suehiro, Takahiro Yamasaki, Isao Sakaida

**Affiliations:** 1Department of Laboratory Science, Graduate School of Medicine, Yamaguchi University, Ube 7558505, Japan; a006upu@yamaguchi-u.ac.jp (S.F.); tkgcrzccg@docomo.ne.jp (T.T.); a003upu@yamaguchi-u.ac.jp (N.K.); otsuyama@yamaguchi-u.ac.jp (K.-i.O.); htsune@yamaguchi-u.ac.jp (H.T.); nojima-j@yamaguchi-u.ac.jp (J.N.); 2Department of Gastroenterology and Hepatology, Graduate School of Medicine, Yamaguchi University, Ube 7558505, Japan; g029ub@yamaguchi-u.ac.jp (S.I.); sakaida@yamaguchi-u.ac.jp (I.S.); 3Department of Oral and Maxillofacial Surgery, Graduate School of Medicine, Yamaguchi University, Ube 7558505, Japan; harako@yamaguchi-u.ac.jp (K.H.); kmishima@yamaguchi-u.ac.jp (K.M.); 4Department of Oncology and Laboratory Medicine, Graduate School of Medicine, Yamaguchi University, Ube 7558505, Japan; ysuehiro@yamaguchi-u.ac.jp (Y.S.); t.yama@yamaguchi-u.ac.jp (T.Y.)

**Keywords:** colorectal cancer, deep ultraviolet, DNA damage, *Fusobacterium nucleatum*, light-emitting diode

## Abstract

Background: *Fusobacterium nucleatum*, which is associated with periodontitis and gingivitis, has been detected in colorectal cancer (CRC). Methods: We evaluated the bactericidal effect of deep ultraviolet (DUV) light-emitting diode (LED) light therapy on *F. nucleatum* both qualitatively and quantitatively. Two DUV-LEDs with peak wavelengths of 265 and 280-nm were used. DNA damage to *F. nucleatum* was evaluated by the production of cyclobutane pyrimidine dimers (CPD) and pyrimidine (6–4) pyrimidone photoproducts (6–4PP). Results: DUV-LEDs showed a bactericidal effect on *F. nucleatum*. No colony growth was observed after 3 min of either 265 nm or 280 nm DUV-LED irradiation. The survival rates of *F. nucleatum* under 265 nm DUV-LED light irradiation dropped to 0.0014% for 10 s and to 0% for 20 s irradiation. Similarly, the survival rate of *F. nucleatum* under 280 nm DUV-LED light irradiation dropped to 0.00044% for 10 s and 0% for 20 s irradiation. The irradiance at the distance of 35 mm from the DUV-LED was 0.265 mW/cm^2^ for the 265 nm LED and 0.415 mW/cm^2^ for the 280 nm LED. Thus, the radiant energy for lethality was 5.3 mJ/cm^2^ for the 265 nm LED and 8.3 mJ/cm^2^ for the 280 nm LED. Amounts of CPD and 6–4PP in *F. nucleatum* irradiated with 265 nm DUV-LED light were 6.548 ng/µg and 1.333 ng/µg, respectively. Conclusions: DUV-LED light exerted a bactericidal effect on *F. nucleatum* by causing the formation of pyrimidine dimers indicative of DNA damage. Thus, DUV-LED light therapy may have the potential to prevent CRC.

## 1. Introduction

The development of gastrointestinal cancer involves lifestyle factors such as alcohol use, smoking [1], obesity [2], and eating habits [3] as well as certain infectious diseases [4]. The International Agency for Research Cancer (IARC) identified *Helicobacter pylori* as a definite oncogenic factor for gastric cancer from epidemiological studies in 1994 [5], and eradication therapy for *H. pylori* has been shown to suppress the development of metachronous gastric cancers [6,7]. Unlike gastric cancer, the mortality rate from colorectal cancer (CRC) has increased significantly in Japan, and thus effective methods to prevent CRC are required.

Gut dysbiosis has been associated with the development of CRC [8]. Recent studies have identified *Fusobacterium nucleatum* (*F. nucleatum*), *Streptococcus bovis*, enterotoxigenic *Bacteroides fragilis*, *Enterococcus faecalis*, *Escherichia coli*, and *Peptostreptococcus anaerobius* as CRC candidate pathogens [9]. *F. nucleatum* has been reported to be detected in CRC [10]. *Fusobacterium* is an anaerobic Gram-negative bacillus present in the oral cavity and digestive tract of healthy individuals [11], and *F. nucleatum* is associated with oral inflammatory diseases such as periodontitis and gingivitis [12,13,14]. In 2012, Kostic et al. characterized the composition of the microbiota in colorectal carcinoma using whole genome sequencing and showed that Fusobacterium sequences were enriched in carcinomas [15]. Castellarin et al. also found that only *F. nucleatum* was significantly increased in CRC tumors relative to control specimens by using RNA sequencing [16]. Mima et al. reported that the amount of *F. nucleatum* DNA in CRC tissue is associated with shorter survival [17], and Yamaoka et al. found that it may potentially serve as a prognostic biomarker [18]. Thus, accumulating evidence suggests that *F. nucleatum* is associated with the development and progression of CRC in humans.

Several studies have revealed a mechanism by which *F. nucleatum* contributes to CRC development. Rubinstein et al. showed that FadA adhesin on *F. nucleatum* binds to E-cadherin and activates β-catenin signaling in CRC cells, and then, CRC growth is induced by transcription of c-Myc and cyclin-D1 [19]. Yang et al. showed that CRC cell lines infected with *F. nucleatum* formed larger tumors more rapidly in nude mice than did uninfected cells. Infection of cells with *F. nucleatum* increased the expression of miR21 by activating TLR4 signaling to MYD88, leading to the activation of nuclear factor NFκB [20]. Yu et al. found that *F. nucleatum* was abundant in CRC tissues of patients with recurrence after chemotherapy. *F. nucleatum* targeted TLR4 and MYD88 innate immune signaling and specific microRNAs to activate the autophagy pathway and alter CRC chemotherapeutic response [21]. Gur et al. found that the Fap2 protein of *F. nucleatum* directly interacted with T-cell immunoglobulin and the ITIM domain (TIGIT), leading to the inhibition of NK cell cytotoxicity [22]. *F. nucleatum* adheres to and invades CRC cells and then induces oncogenic and inflammatory responses to stimulate their growth.

We have considered deep ultraviolet (DUV) light-emitting diode (LED) light therapy for the treatment of *F. nucleatum*. Because *F. nucleatum* forms a biofilm in the periodontal pocket, injection of an antibacterial agent does not provide a sufficient bactericidal effect, and infection often recurs [23]. LEDs are semiconductor devices that emit light and are ideal for downsizing equipment [24]. Recently, LEDs that can emit ultraviolet light have been developed [25], and we have confirmed that DUV-LEDs have various bactericidal effects against bacteria and fungi in vitro [26]. In the present study, the bactericidal effect of DUV-LEDs on *F. nucleatum* was examined.

## 2. Materials and Methods

### 2.1. Fusobacterium nucleatum Strain and Growth Conditions

*F. nucleatum* (ATCC 25586) and *Escherichia coli* (NBRC3972) were purchased from NITE Biological Resource Center (NBRC). The frozen bacterial stock of *F. nucleatum* was cultivated in Gifu Anaerobic Medium (GAM) agar in an anaeropack at 37 °C for 72 h. *E. coli* were cultured on heart infusion agar (Eiken Kagaku Co., Ltd., Tokyo, Japan) at 37 °C for 16 h. Then, cell suspensions were prepared using phosphate-buffered saline (PBS).

### 2.2. Irradiation by DUV-LEDs

Two DUV-LEDs with peak wavelengths of 265 and 280-nm (VPS131 (265-nm LED) and VPS161 (280-nm LED), NIKKISO CO., LTD., Tokyo, Japan) were used (Figure 1). Total radiant flux of the DUV-LEDs was 9.4 mW for the 265 nm and 17.0 mW for the 280 nm wavelength. As the half-power angle of the LEDs was 130 degrees, we set the distance between the LED and the plate at 35 mm, which enabled uniform DUV-LED illumination (Figure 2). The rated current for the 265 nm LEDs set by the company is 350 mA. We used a direct current-regulated power supply in constant current mode (PW36-1.5AD, TEXIO TECHNOLOGY CORPORATION, Yokohama, Japan). The current was set to 350 mA or 50 mA to irradiate LED light. Irradiance was measured with an MCPD-9800 array spectrometer (Otsuka Electronics, Osaka, Japan).

### 2.3. Qualitative Test of Bactericidal Effect on Agar Medium Surface by DUV-LED Light

The suspension of *F. nucleatum* was adjusted to 1 McFarland turbidity standard with PBS and then spread on GAM agar plates (diameter, 35 mm) with a cotton swab. We conducted time series tests with irradiation times of 0, 10, 20, 30, 60, and 180 s. Colonies of *F. nucleatum* were observed after anaerobic cultivation for 6 days. As the non-irradiated control, *F. nucleatum* was left in the air for the same time periods as the DUV-LED irradiation.

### 2.4. Quantitative Test of Bactericidal Effect of DUV-LED Light

Bactericidal effects were determined using a colony-forming assay. The cell suspension of *F. nucleatum* was adjusted to 1.0 × 10^6^ colony forming units (CFU)/mL. One milliliter of cell suspension was dispensed onto a 35 mm dish and irradiated by DUV-LED light. We conducted time series tests with irradiation times of 0, 10, 20, and 30 s. Then, 10 μL of 10-fold serial dilutions of the cell suspension was plated onto GAM agar in quadruplicate, and the plates were incubated at 37 °C for 6 days. Thereafter, the number of colonies was counted, and survival rate was expressed as a percentage of the non-irradiated control. The experiment was performed three times, and the average percentage of surviving *F. nucleatum* was evaluated [27,28].

### 2.5. Cytotoxicity of Human Keratinocytes by DUV-LED

HaCaT cells, a spontaneously immortalized human keratinocyte line, were cultured in Dulbecco’s Modified Eagle Medium—high glucose (4.5 g/L) with L-glutamine and with sodium pyruvate (Capricorn) supplemented with 10% fetal bovine serum. HaCaT cells (4.0 × 10^5^/dish) were seeded in 35 mm dishes and cultured at 37 °C in 5% CO_2_ for 24 h. The cells were then washed twice with PBS, after which we immediately irradiated the cells with DUV-LED light. Two milliliters of fresh medium was added, and the mixture was cultured for 24 h. Then, the cells were harvested, and cell viability was evaluated by trypan blue dye exclusion assay. The number of viable cells was counted, and the survival rate was expressed as a percentage of the non-irradiated control. The experiment was performed three times, and the average percentage of surviving HaCaT cells was evaluated.

### 2.6. CPD and 6–4PP Quantification by ELISA

The DNA damage in *F. nucleatum* was evaluated based on the production of cyclobutane pyrimidine dimers (CPD) and pyrimidine (6–4) pyrimidone photoproducts (6–4PP) with an OxiSelect™ UV-Induced DNA Damage ELISA Kit (CELL BIOLABS Inc., San Diego, CA, USA). The bacterial suspension was adjusted to a 2 McFarland turbidity standard. Then, 2 mL of the suspension was dispensed onto a 35 mm dish, followed by DUV-LED light irradiation for 30 s. DNA was extracted from 1.8 mL of the bacterial solutions with a QIAamp DNA FFPE Tissue Kit (QIAGEN, Venlo, The Netherlands). The ELISA procedure was performed according to the manufacturer’s protocol. We used DNA samples from *E. coli* irradiated by DUV from a UV lamp as a positive control and DNA samples from *F. nucleatum* cultured without DUV irradiation as a negative control. The UV lamp (GL-15, Toshiba, Japan) was installed in a safety cabinet (Panasonic Healthcare, Tokyo, Japan).

### 2.7. Statistics

Differences in survival rates between the 265 nm DUV-LED and 280 nm DUV-LED were analyzed by *t*-test using StatFlex Ver.6.0 (Artech Co., Ltd., Osaka, Japan).

## 3. Results

### 3.1. Qualitative Test of Bactericidal Effect on Agar Medium Surface by DUV-LED Light

DUV-LEDs with peak wavelengths of 265 and 280-nm showed similar bactericidal effect on *F. nucleatum* cultured on an agar medium surface in a time-dependent manner. No colony growth was observed after 180 s of either 265 nm or 280 nm DUV-LED irradiation (Figure 3). In this experiment, even if the sterilized dish was returned to the culture, no bacterial growth was observed. We confirmed that an increase in the ambient temperature was not observed when we supplied a current of 350 mA to the DUV-LEDs.

### 3.2. Quantitative Test of Bactericidal Effect by DUV-LED Light

When we supplied a current of 350 mA to the DUV-LED, the survival rate of *F. nucleatum* under 265 nm DUV-LED light irradiation dropped to 0.0014% for 10 s and to 0% (below limit of detection) for 20 s irradiation. Similarly, the survival rate of *F. nucleatum* under 280 nm DUV-LED light irradiation dropped to 0.00044% for 10 s and 0% (below limit of detection) for 20 s irradiation (Figure 4). When 50 mA was supplied to the DUV-LED, the survival rates of *F. nucleatum* under 265 nm DUV-LED light irradiation were 45.34% for 10 s and 16.13% for 20 s irradiation. Whereas those of *F. nucleatum* under 280 nm DUV-LED light irradiation were 54.72% for 10 s and 21.34% for 20 s irradiation (Figure 4). There was no significant difference in the survival rates of *F. nucleatum* between the 265 nm DUV-LED and 280 nm DUV-LED irradiations. To confirm the presence of any surviving bacteria, 0.5 mL of these residual suspensions was cultured in liquid medium, but no *F. nucleatum* growth was observed.

When we supplied a current of 350 mA to the DUV-LED, the irradiance at the distance of 35 mm from the DUV-LED was 0.265 mW/cm^2^ for the 265 nm LED and 0.415 mW/cm^2^ for the 280 nm LED. The irradiation time under 265 nm and 280 nm DUV-LED light for which the survival rates were 0% was 20 s. Thus, the radiant energy for lethality was 5.3 mJ/cm^2^ for the 265 nm LED and 8.3 mJ/cm^2^ for the 280 nm LED.

### 3.3. Evaluation of HaCaT Cell Damage by DUV-LED Light

When we supplied a current of 350 mA to the DUV-LED, the survival rates of HaCaT cells under 265 nm DUV-LED light irradiation were 4.38% for 10 s and 2.08% for 20 s irradiation. Similarly, the survival rates of HaCaT cells under 280 nm DUV-LED light irradiation dropped to 3.31% for 10 s and 2.26% for 20 s irradiation (Figure 5). When 50 mA was supplied to the DUV-LED, the survival rates of HaCaT cells under 265 nm DUV-LED light irradiation were 90.14% for 10 s and 48.29% for 20 s irradiation, whereas those of HaCaT cells under 280 nm DUV-LED light irradiation were 90.06% for 10 s and 47.76% for 20 s irradiation (Figure 5). There was no significant difference in the survival rates of HaCaT cells between the 265 nm and 280 nm DUV-LED irradiations.

### 3.4. Detection of DUV-LED Light-Induced DNA Damage by Formation of CPD and 6–4PP

CPD and 6–4PP were detected by ELISA in the DNA samples from *F. nucleatum* irradiated with DUV-LED light. The respective amounts of CPD and 6–4PP in *F. nucleatum* irradiated with 265 nm DUV-LED were 6.548 ng/µg and 1.333 ng/µg, whereas those in *F. nucleatum* irradiated with 280 nm DUV-LED were 7.963 ng/µg and 1.593 ng/µg. As a control, we prepared *E. coli* irradiated with the UV lamp. The respective amounts of CPD and 6–4PP in the *E. coli* were 2.622 ng/µg and 0.276 ng/µg (Figure 6).

## 4. Discussion

Light irradiation from DUV-LEDs exerted a bactericidal effect on *F. nucleatum* in a time-dependent manner. We previously reported that the Gram-negative bacteria *Pseudomonas aeruginosa* and *E. coli* were highly sensitive to DUV-LED light [28]. *F. nucleatum* is an anaerobic Gram-negative bacterium with a similar sensitivity to DUV-LED as that of other Gram-negative bacteria because their cell wall is thinner than that of Gram-positive bacteria [29]. That we did not conduct experiments to form biofilms in vitro is a limitation of the present study.

DUV light at a wavelength of around 260 nm is absorbed by DNA [30]. When cells are irradiated with UV light, covalent bonds of the pyrimidine base form with the adjacent thymine or cytosine base that result in the creation of pyrimidine dimers. These dimers interfere with DNA replication and transcription that can lead to cell death, mutation, and chromosomal instability [31]. The mechanism of the bactericidal effect exerted by DUV-LEDs is thought to be due to DNA damage because DNA extracted from DUV-LED-irradiated *F. nucleatum* contains pyrimidine dimers. While the light wavelength around 280 nm is absorbed by proteins, the effect on proteins of cell membranes is one of the issues being pursued.

Although fusobacteria are found in CRC [10,15,16,17,18], the mechanisms by which they hone in on and localize to CRC have been underexplored. Abed et al. suggested that fusobacterial Fap2 and host Gal-GalNAc are involved in fusobacterial CRC localization and enrichment. They showed that intravenously injected *F. nucleatum* localizes to mouse tumor tissues in a Fap2-dependent manner, suggesting that fusobacteria use a hematogenous route to reach colon adenocarcinomas [32]. Komiya et al. reported that 75% (6/8) of CRC patients exhibited identical strains of *F. nucleatum* in their CRC and saliva specimens [33]. In addition, transient bacteremia is common during periodontal disease with bacterial loads reaching 10^4^ bacteria/mL blood 15 min after tooth brushing in humans [34]. These findings suggest that *F. nucleatum* in CRC originates in the oral cavity. We think that sterilizing *F. nucleatum* in the oral cavity by DUV-LED light may reduce the number of *F. nucleatum* in the intestine and could potentially lead to the prevention of CRC. In addition, we previously showed that stage IV CRC patients with a high *F. nucleatum* copy number had a significantly shorter overall survival time than those with a low copy number [18], indicating that DUV-LED therapy may also improve survival in patients with advanced CRC.

DNA damage by DUV is elicited not only in microorganisms but also in human cells, and chronic exposure to DUV light has been established as a human health hazard [35]. In a mouse model, significant hyperplasia and intercellular edema were induced in the epidermis after chronic irradiation of 254 nm DUV at 4500 mJ/cm^2^ [36]. The radiant energy required for lethality of *F. nucleatum* in the present study was much less, 5.3 mJ/cm^2^ for the 265 nm LED and 8.3 mJ/cm^2^ for the 280 nm LED. By reducing the current supplied to the LEDs, the survival rate of HaCaT cells increased. We must first determine appropriate doses of DUV that can selectively inactivate *F. nucleatum* in an animal model.

DUV-LED light therapy is not a specific treatment targeting one pathogen such as vaccines do, so it may disturb the oral bacterial flora. Treatments that affect all oral bacteria can lead to nitric oxide pathways and adversely affect blood pressure regulation [37]. Thus, we must be aware of these adverse events. In the case of general oral care, however, the number of bacteria is reduced by 60 to 70%, but this is effective in preventing aspiration pneumonia [38]. Consequently, it is considered clinically useful to reduce multiple types of bacteria as a whole rather than suppressing just one type of bacteria. We believe that this also needs to be verified by animal experiments.

LEDs have unique characteristics of compactness, durability, and low heat production, and thus, DUV-LEDs can be applied in narrow spaces such the oral cavity. A mouthpiece-type phototherapy device should be considered as an effective application to irradiate DUV-LED light to a localized infection. It might be necessary to use a thin fiber to irradiate DUV into the periodontal pockets.

## 5. Conclusions

Irradiation with DUV-LED light exerted a bactericidal effect on *F. nucleatum* by causing the formation of pyrimidine dimers indicative of DNA damage. DUV-LED devices can be useful tools for the inactivation of *F. nucleatum*. Thus, DUV-LED light therapy may have the potential to prevent CRC.

## Figures and Tables

**Figure 1 microorganisms-09-00430-f001:**
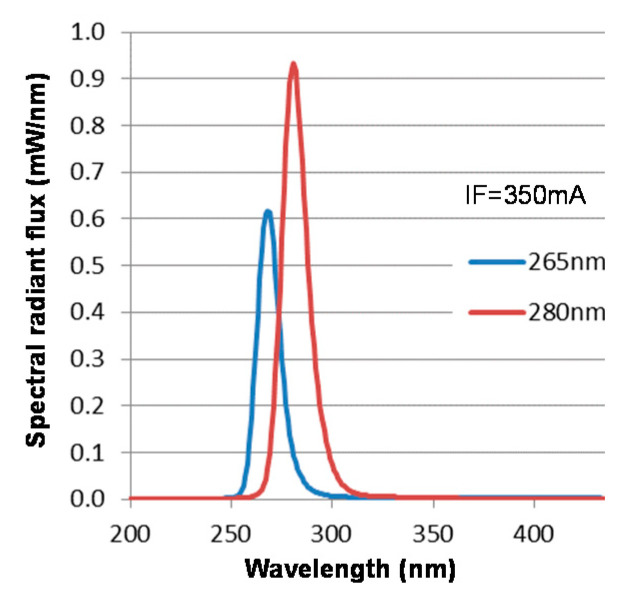
Spectral radiant flux of the DUV-LEDs.

**Figure 2 microorganisms-09-00430-f002:**
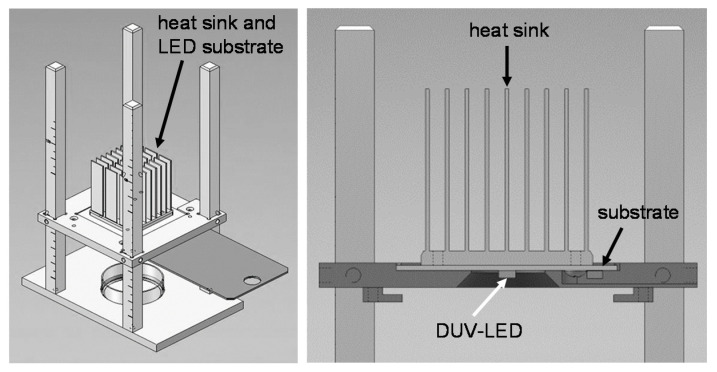
The apparatus for DUV-LED irradiation. The size of the DUV-LEDs was 3.5 mm × 3.5 mm. The size of the substrate of the LEDs was 35 mm × 35 mm × 5 mm. The length of the fins of the heat sink was 32 mm.

**Figure 3 microorganisms-09-00430-f003:**
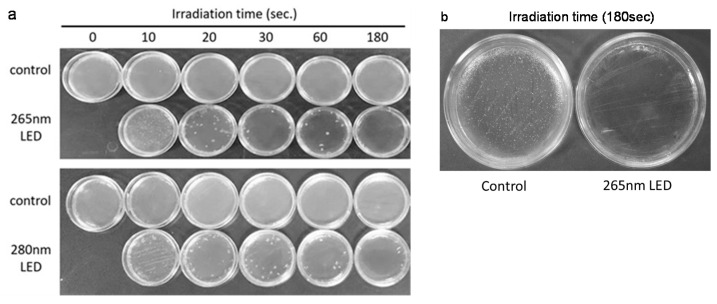
Qualitative test of the bactericidal effect of DUV-LED light irradiation on *Fusobacterium nucleatum*. (**a**) The growth of colonies of *F. nucleatum* was observed after irradiation with 265 nm LED and 280 nm LED at each time point. A non-irradiated control was used. (**b**) An enlarged image of the agar plates.

**Figure 4 microorganisms-09-00430-f004:**
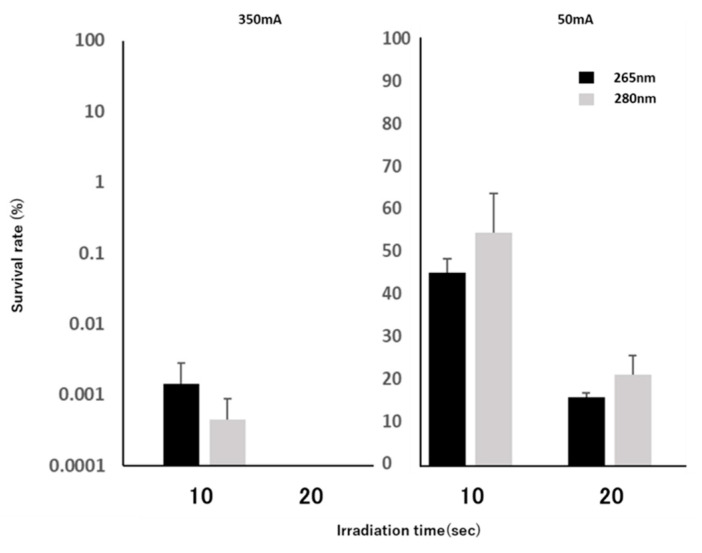
Quantitative test of the bactericidal effect of DUV-LED light irradiation on *Fusobacterium nucleatum*. The survival rates of *F. nucleatum* after irradiation with 265 nm LED (black bars) and 280 nm LED (grey bars) are shown for the application of 350 mA or 50 mA of current. Error bars indicate standard error.

**Figure 5 microorganisms-09-00430-f005:**
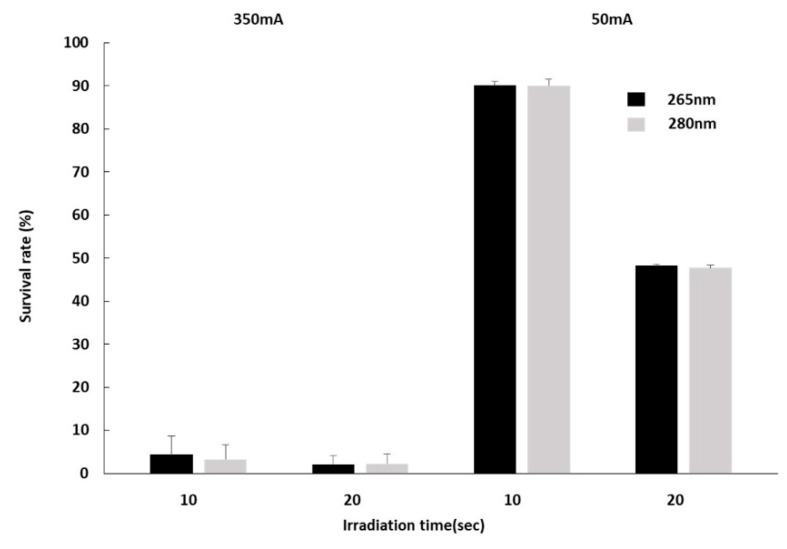
Evaluation of HaCaT cell damage by DUV-LED light. The survival rates of HaCaT cells after irradiation with 265 nm LED (black bars) and 280 nm LED (grey bars) are shown for the application of 350 mA or 50 mA of current. Error bars indicate standard error.

**Figure 6 microorganisms-09-00430-f006:**
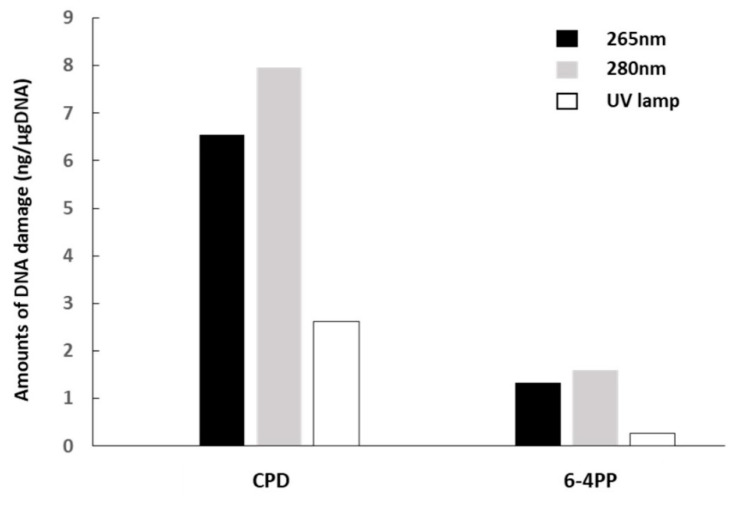
Detection of DUV-LED light-induced DNA damage by quantification of CPD and 6–4PP with ELISA. DNA samples extracted from *F. nucleatum* after 30 s irradiation with 265 nm LED light (black bars) or 280 nm LED light (grey bars) were measured for amounts of CPD and 6–4PP per microgram of DNA. *E. coli* irradiated with a UV lamp (hatched bar) were used as a positive control.

## Data Availability

All data generated or analyzed during the study are included in this published article.

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
