# Peer review of "Deep Ultraviolet Light-Emitting Diode Light Therapy for Fusobacterium nucleatum"

_microorganisms, 2021, doi:10.3390/microorganisms9020430_

Round 1
Reviewer 1 Report
Dear Editor,
Subject matter is interesting. However, there are concerns regarding the rational of using UV light, a known mutagenic to mammalian cells especially when other safer wavelengths (i.e. blue light) were shown to be effective. Results are poorly presented and seems preliminary and lacking. Sample size is small; the phototoxic effect was measured only on DNA whereas the effect on other biomolecules such as lipids and proteins (e.g. membrane integrity) were not monitored. Light distance and exposure times were not justified and heat production was not monitored. It is unclear why DNA damage was not studied following the various exposure times (only 30s was shown). Therefore, in its current form the MS contribution to the field is questionable.
Author Response
Reviewer 1
Subject matter is interesting. However, there are concerns regarding the rational of using UV light, a known mutagenic to mammalian cells especially when other safer wavelengths (i.e. blue light) were shown to be effective. Results are poorly presented and seems preliminary and lacking. Sample size is small; the phototoxic effect was measured only on DNA whereas the effect on other biomolecules such as lipids and proteins (e.g. membrane integrity) were not monitored.
Answer) As we previously investigated the effect of the 405-nm LED on bacteria and found weaker bactericidal effect as compared to DUV-LEDs, we focused on DUV-LEDs. As reviewer 1 mentions, the light wavelength around 280 nm is absorbed by proteins, and the effect on proteins of cell membranes is one of the issues being pursued. We will perform additional study to resolve this issue in the near future.
Light distance and exposure times were not justified and heat production was not monitored.
Answer) We fixed the distance between the LED and the plate at 35 mm, which enabled uniform illumination by the LED because the half power angle of the LEDs was 130 degrees shown in Table 1. We conducted time series tests with exposure times of 0, 10, 20, 30, 60, and 180 s. We used the heat sink for the LEDs shown in the Figure 2 and confirmed that an increase in the ambient temperature was not observed. The output wavelength and radiant flux of the LEDs did not change for at least 300 s when we supplied a current of 350 mA to the DUV-LEDs.
(method) As the half power angle of the LEDs was 130 degrees, we set the distance between the LED and the plate at 35 mm which enabled uniform DUV-LED illumination.
(result) We confirmed that an increase in the ambient temperature was not observed when we supplied a current of 350 mA to the DUV-LEDs.
It is unclear why DNA damage was not studied following the various exposure times (only 30s was shown). Therefore, in its current form the MS contribution to the field is questionable.
Answer) We performed time-series tests to evaluate bactericidal effects of DUV-LED irradiation on F. nucleatum and eventually found that 30 s of exposure time has the best bactericidal effect. Thus, we investigated the DNA damage only at 30 s of exposure time.
We admit that the current study is preliminary because the sample size was small, and only DNA damage was investigated as a mechanism of sterilization. However, we believe our study has the potential to help prevent CRC and it is worth publishing in the special issue “The Oral Microbiome in Systemic Health and Disease: Therapeutics for Reversing Microbial Dysbiosis and Applications in Precision Medicine” of Microorganisms.
Reviewer 2 Report
In the abstract, include the power density used for irradiation (in Watts/cm2) and include the radiant energy for lethality as this is critical information.
In the start of the introduction, mention the role of gut dysbiosis and other key pathogens such as enterotoxigenic Bacteroides fragilis, and cite current research on the microbiome and colorectal cancer, to give a wider view of its importance.
There is no need to mention hepatocellular carcinoma and hepatitis B and C viruses as this is extraneous to the topis of the paper.
In the methods section, the nfollowing information is missing.
State the spectral bandwidth of the LEDs used (from the manufacturer's specifications).
Provide the Nikkiso product code or description e.g. VPS131, PearlBeam etc) - the level of detail must be sufficient to allow others to reproduce the study.
State in degrees the directional half-power angle, as this explains the light distribution.
Including a diagram or photo of the actual setup used for irradiation would be very useful to show the distribution of light across the plates.
As LED output is determined by current, explain how the drive current values were selected and optimised.
State what power supply was used, and was it a constant current type?
As LED output wavelength is influenced by device temperature, explain any heatsinking arrangements used.
Was irradiation conducted with the lids removed from the plates, or not?
What dilution series were used for the plate counts?
Results:
In Figure 1, there should be an untreated (non-irradiated) control showing obvious colonies in the upper part of each panel. The images supplied do not show these.
Discussion:
As F. nucleatum is a normal oral commensal organism, it would be more logical to discuss its suppression (or elimination) rather than to use the term "sterilizing".
The authors should discuss what other options would there be for elimination that would be more specific and not disrupt the oral microbiome (e.g. vaccination). UVC treatment in the mouth using a mouthguard would influence all bacteria and this could have wider consequences that are unfavourable - such as disturbing the nitrate-NO pathway and altering blood pressure regulation. Such concerns should be noted.
Given the location of F. nucleatum deep in dental plaque biofilms and in subgingival areas, the authors need to discuss the optical problems of delivering UVC light into confined areas (e.g. problems of UVC light transmission), as well as safety concerns for the oral soft tissues as these are important issues for translating this research into practice.
Author Response
Reviewer 2
In the abstract, include the power density used for irradiation (in Watts/cm2) and include the radiant energy for lethality as this is critical information.
Answer) We added the results on the power density of the LEDs and the radiant energy required for lethality in the abstract.
(abstract) The irradiance at the distance of 35 mm from the DUV-LED was 0.265 mW/cm2 for the 265-nm LED and 0.415 mW/cm2 for the 280-nm LED. Thus, the radiant energy for lethality was 5.3 mJ/cm2 for the 265-nm LED and 8.3 mJ/cm2 for the 280-nm LED.
In the start of the introduction, mention the role of gut dysbiosis and other key pathogens such as enterotoxigenic Bacteroides fragilis, and cite current research on the microbiome and colorectal cancer, to give a wider view of its importance.
Answer) Thank you for this suggestion. We added the explanation about gut dysbiosis and CRC and reference articles.
(introduction) Gut dysbiosis has been associated with the development of colorectal cancer (CRC). Recent studies have identified Fusobacterium nucleatum, Streptococcus bovis, enterotoxigenic Bacteroides fragilis, Enterococcus faecalis, Escherichia coli, and Peptostreptococcus anaerobius as CRC candidate pathogens.
(references)
- Chattopadhyay I, Dhar R, Pethusamy K, Seethy A, Srivastava T, Sah R, Sharma J, Karmakar S. Exploring the Role of Gut Microbiome in Colon Cancer. Appl Biochem Biotechnol. 2021 Jan 25. doi: 10.1007/s12010-021-03498-9.
- Cheng Y, Ling Z, Li L. The Intestinal Microbiota and Colorectal Cancer. Front Immunol. 2020 Nov 30;11:615056. doi: 10.3389/fimmu.2020.615056.
There is no need to mention hepatocellular carcinoma and hepatitis B and C viruses as this is extraneous to the topis of the paper.
Answer) Thank you for this suggestion. We deleted the description about hepatocellular carcinoma.
In the methods section, the following information is missing. State the spectral bandwidth of the LEDs used (from the manufacturer's specifications). Provide the Nikkiso product code or description e.g. VPS131, PearlBeam etc) - the level of detail must be sufficient to allow others to reproduce the study. State in degrees the directional half-power angle, as this explains the light distribution. Including a diagram or photo of the actual setup used for irradiation would be very useful to show the distribution of light across the plates.
A) We added information in the methods section as you recommended. We showed the spectral data of the LEDs in Figure 1 and provide sufficient information on the LED in the method. Their product codes were VPS131 and VPS161. The half power angle of both LEDs was 130 degrees. The rated current for the 265-nm LED set by the company is 350 mA. We now show our apparatus for the experiments in Figure 2.
As LED output is determined by current, explain how the drive current values were selected and optimised.
Answer) The rated current for the 265-nm LEDs set by the company is 350 mA.
State what power supply was used, and was it a constant current type?
Answer) We used a direct current (DC)-regulated power supply in constant current mode (PW36-1, TEXIO TECHNOLOGY CORPORATION, Yokohama, Japan).
As LED output wavelength is influenced by device temperature, explain any heatsinking arrangements used.
Answer) We used the heat sink which length of the fins was 32mm shown in Figure 2 and confirmed that an increase in the ambient temperature was not observed. The output wavelength and the radiant flux of the LEDs did not change for at least 300 s when we supplied a current of 350 mA to the DUV-LEDs.
Was irradiation conducted with the lids removed from the plates, or not?
What dilution series were used for the plate counts?
Answer) We removed the lids from the plates during DUV-LED irradiation. The cell suspension of F. nucleatum was adjusted to 1.0×106 colony forming units (CFU)/mL for the colony forming assay.
Results:
In Figure 1, there should be an untreated (non-irradiated) control showing obvious colonies in the upper part of each panel. The images supplied do not show these.
Answer) We added an enlarged image of the agar plates.
Discussion:
As F. nucleatum is a normal oral commensal organism, it would be more logical to discuss its suppression (or elimination) rather than to use the term "sterilizing".
The authors should discuss what other options would there be for elimination that would be more specific and not disrupt the oral microbiome (e.g. vaccination). UVC treatment in the mouth using a mouthguard would influence all bacteria and this could have wider consequences that are unfavourable - such as disturbing the nitrate-NO pathway and altering blood pressure regulation. Such concerns should be noted.
Answer) Thank you for your valuable comments. As you mentioned, DUV-LED light therapy is not a specific treatment targeting one pathogen such as vaccines do, so it may disturb the oral bacterial flora. Treatments that affect all oral bacteria can lead to nitric oxide pathways and adversely affect blood pressure regulation. Thus, we must be aware of these adverse events. In the case of general oral care, however, the number of bacteria is reduced by 60 to 70%, but this is effective in preventing aspiration pneumonia. Consequently, it is considered clinically useful to reduce multiple types of bacteria as a whole rather than suppressing just one type of bacteria. We believe that this also needs to be verified by animal experiments.
(discussion) DUV-LED light therapy is not a specific treatment targeting one pathogen such as vaccines do, so it may disturb the oral bacterial flora. Treatments that affect all oral bacteria can lead to nitric oxide pathways and adversely affect blood pressure regulation. Thus, we must be aware of these adverse events. In the case of general oral care, however, the number of bacteria is reduced by 60 to 70%, but this is effective in preventing aspiration pneumonia. Consequently, it is considered clinically useful to reduce multiple types of bacteria as a whole rather than suppressing just one type of bacteria. We believe that this also needs to be verified by animal experiments.
(reference)
Barbadoro P, Ponzio E, Coccia E, Prospero E, Santarelli A, Rappelli GGL, D'Errico MM. Association between hypertension, oral microbiome and salivary nitric oxide: A case-control study. Nitric Oxide. 2021 Jan1;106:66-71. doi: 10.1016/j.niox.2020.11.002.
Yoneyama T, Yoshida M, Matsui T, Sasaki H. Oral care and pneumonia. Oral Care Working Group. Lancet. 1999 Aug 7;354(9177):515. doi: 10.1016/s0140-6736(05)75550-1.
Given the location of F. nucleatum deep in dental plaque biofilms and in subgingival areas, the authors need to discuss the optical problems of delivering UVC light into confined areas (e.g. problems of UVC light transmission), as well as safety concerns for the oral soft tissues as these are important issues for translating this research into practice.
Answer) As you pointed out, we think that it is necessary to use a thin fiber to irradiate DUV into the periodontal pockets. As we mentioned in the discussion, F. nucleatum can be suppressed with much less energy than the amounts of UV irradiation that affect the human body, but safety still needs to be evaluated and verified using an animal model. In addition, we plan to study the effects of 222-nm DUV, which does not affect the human body and has a bactericidal effect, in the future.
(discussion) It might be necessary to use a thin fiber to irradiate DUV into the periodontal pockets.
Reviewer 3 Report
The manuscript "Deep Ultraviolet Light-emitting Diode Light Therapy for Fusobacterium Nucleatum" describes the survival and DNA damage to F. nucleatum using wavelengths 265nm and 280nm. The manuscript is clear, concise, interesting and well written. I do have a few recommendations for how to improve the paper and some questions for the author which need to be addressed.
Minor comments:
line 142-143 states that no colony growth was observed at wavelengths 265nm or 280nm yet Figure 1 shows some colonies observed at 180 seconds at 280nm.
A better description of the DUV-LED device and modules would aid readers in understanding how the UV was administered.
Major comments:
The authors expose F. nucleatum to DUV-LED either on an agar surface or in a liquid cell suspension in a 35mm dish yet in both instances correction factors are not reported such as, for example, Petri factor (indicating the area of even distribution of irradiated light on a Petri dish), reflectance factor (degree of UV light reflected from a surface) and absorption coefficient (measurement of how strongly a substance e.g. liquid absorbs light). These correction factors would give the readers a better idea of how effective the UV dose is in penetrating substances and reaching bacteria.
The authors acknowledge that a limitation of their study was that they did not perform biofilm experiments when conducting DUV-LED experiments (line 204-205). However there may be a second limitation. Many bacteria have repair mechanisms to remove CPD and other DNA damage caused by UV (photoreactivation). While this was not tested in this study, the possibility of it occurring should be acknowledged and discussed in the manuscript.
Author Response
Reviewer 3
The manuscript "Deep Ultraviolet Light-emitting Diode Light Therapy for Fusobacterium Nucleatum" describes the survival and DNA damage to F. nucleatum using wavelengths 265nm and 280nm. The manuscript is clear, concise, interesting and well written. I do have a few recommendations for how to improve the paper and some questions for the author which need to be addressed.
- Minor comments:
line 142-143 states that no colony growth was observed at wavelengths 265nm or 280nm yet Figure 1 shows some colonies observed at 180 seconds at 280nm.
Answer) These were wounds on the agar medium, not colonies. We added an enlarged image of the agar plates.
- A better description of the DUV-LED device and modules would aid readers in understanding how the UV was administered.
Answer) We now show the apparatus used for the experiments in Figure 2.
- Major comments:
The authors expose F. nucleatum to DUV-LED either on an agar surface or in a liquid cell suspension in a 35mm dish yet in both instances correction factors are not reported such as, for example, Petri factor (indicating the area of even distribution of irradiated light on a Petri dish), reflectance factor (degree of UV light reflected from a surface) and absorption coefficient (measurement of how strongly a substance e.g. liquid absorbs light). These correction factors would give the readers a better idea of how effective the UV dose is in penetrating substances and reaching bacteria.
Answer) We fixed the distance between the LED and the plate at 35 mm, which enabled uniform illumination of the LED because the half power angle of the LEDs was 130 degrees. The cell suspension of F. nucleatum was adjusted to 1.0×106 CFU/mL by PBS. We used 1 mL of cell suspension dispensed onto a 35-mm dish for irradiation by DUV-LED light for the colony forming assay. As the bacterial solution is adjusted with colorless and transparent PBS and the amount of the solution is very small, we considered that DUV light is hardly absorbed and thus can reach the bacteria. We do not know how to evaluate the reflectance factor.
(method) As the half power angle of the LEDs was 130 degrees, we set the distance between the LED and the plate at 35 mm which enabled uniform DUV-LED illumination.
- The authors acknowledge that a limitation of their study was that they did not perform biofilm experiments when conducting DUV-LED experiments (line 204-205). However there may be a second limitation. Many bacteria have repair mechanisms to remove CPD and other DNA damage caused by UV (photoreactivation). While this was not tested in this study, the possibility of it occurring should be acknowledged and discussed in the manuscript.
Answer) As you mention, many bacteria have repair mechanisms to remove CPD and other DNA damage caused by UV. In the experiments, even if the sterilized dish was returned to the culture, no bacterial growth was observed.
(result) In this experiment, even if the sterilized dish was returned to the culture, no bacterial growth was observed.
Round 2
Reviewer 1 Report
The authors stated in their reply that "We admit that the current study is preliminary". This reviewer is in total agreement with that statement and doesn't feel that in its current form it is suitable for publication. The authors also stated that "the effect on proteins of cell membranes is one of the issues being pursued. We will perform additional study to resolve this issue in the near future." Once these additional experiments will be conducted and additional data provided it might be suitable. However, the authors are also encouraged to take notice of the major criticism regarding methodology.
Reviewer 2 Report
All my points have been addressed and I am happy for the paper to proceed to publication.
Reviewer 3 Report
I am happy with the revised manuscript and recommend it for publication.